# Mutation of GGMP Repeat Segments of *Plasmodium falciparum* Hsp70-1 Compromises Chaperone Function and Hop Co-Chaperone Binding

**DOI:** 10.3390/ijms22042226

**Published:** 2021-02-23

**Authors:** Stanley Makumire, Tendamudzimu Harmfree Dongola, Graham Chakafana, Lufuno Tshikonwane, Cecilia Tshikani Chauke, Tarushai Maharaj, Tawanda Zininga, Addmore Shonhai

**Affiliations:** 1Department of Biochemistry, University of Venda, Private Bag X5050, Thohoyandou 0950, South Africa; stanley.makumire@uct.ac.za (S.M.); tendadongola@gmail.com (T.H.D.); graham.chakafana@uct.ac.za (G.C.); lufunochiko@gmail.com (L.T.); tshikanicecilia@gmail.com (C.T.C.); tzininga@sun.ac.za (T.Z.); 2Structural Biology Research Unit, Department of Integrative Biomedical Sciences, University of Cape Town, Observatory 7925, South Africa; 3Department of Medicine, University of Cape Town, Faculty of Health Sciences, Observatory, Cape Town 7925, South Africa; 4Department of Biochemistry, Stellenbosch University, Stellenbosch 7600, South Africa; 20222483@sun.ac.za

**Keywords:** malaria, *Plasmodium falciparum*, chaperone, GGMP repeats, Hsp70, Hop

## Abstract

Parasitic organisms especially those of the Apicomplexan phylum, harbour a cytosol localised canonical Hsp70 chaperone. One of the defining features of this protein is the presence of GGMP repeat residues sandwiched between α-helical lid and C-terminal EEVD motif. The role of the GGMP repeats of Hsp70s remains unknown. In the current study, we introduced GGMP mutations in the cytosol localised Hsp70-1 of *Plasmodium falciparum* (PfHsp70-1) and a chimeric protein (KPf), constituted by the ATPase domain of *E. coli* DnaK fused to the C-terminal substrate binding domain of PfHsp70-1. A complementation assay conducted using *E. coli dnaK756* cells demonstrated that the GGMP motif was essential for chaperone function of the chimeric protein, KPf. Interestingly, insertion of GGMP motif of PfHsp70-1 into DnaK led to a lethal phenotype in *E. coli dnaK756* cells exposed to elevated growth temperature. Using biochemical and biophysical assays, we established that the GGMP motif accounts for the elevated basal ATPase activity of PfHsp70-1. Furthermore, we demonstrated that this motif is important for interaction of the chaperone with peptide substrate and a co-chaperone, PfHop. Our findings suggest that the GGMP may account for both the specialised chaperone function and reportedly high catalytic efficiency of PfHsp70-1.

## 1. Introduction

The main agent of malaria, *Plasmodium falciparum*, expresses 6 members of the heat shock protein 70 (Hsp70) family. Of these, *P. falciparum* Hsp70-1 (PfHsp70-1) and PfHsp70-z localise to the parasite cytosol [1,2]. PfHsp70-1 is implicated in parasite cyto-protection through its role as a protein folding facilitator [3]. Small molecule Hsp70 inhibitors which exhibit anti-plasmodial activity have been identified [4,5] and some of these appear to selectively target the function of parasite Hsp70 with minimum adverse effects on the chaperone function of human Hsp70 [6]. This suggests that targeting the essential Hsp70 function represents a potential avenue in the design of alternative antimalarial therapies [7,8]. In addition, heat shock proteins of the parasite, amongst them, Hsp70 are implicated in parasite resistance to antimalarial drugs, including the currently used first line treatment, artemisinin combination therapies [9]. This has raised prospects for circumventing antimalarial drug resistance by targeting heat shock protein function in antimalarial combination therapies [8,10]. Indeed, recent work on structural elucidation of *P. falciparum* Hsp70 and its co-chaperones [11,12] immensely contributes towards targeting of these proteins in antimalarial drug discovery.

One of the most intriguing aspects of members of the Hsp70 family of chaperones is that despite their general sequence conservation, they exhibit functional specificity [6,13]. The identification of the unique structure-function features of Hsp70 is therefore a priority in this regard. Hsp70 is made up of two domains: the N-terminal ATPase domain and a C-terminal substrate binding domain (SBD). The two domains are linked by a highly conserved linker. While the ATPase domain of Hsp70 is highly conserved, the SBD of Hsp70 is more divergent. Since the ATPase domain of Hsp70 is more conserved than its SBD, the functional specificity of the protein is largely attributed to the SBD. However, adding to the intrigue is that the function of Hsp70 is thought to be regulated at least to a large degree by its co-chaperones [14]. These include so-called J proteins/Hsp40 co-chaperones whose primary binding site is in the ATPase domain of Hsp70 [15]. Hsp40 co-chaperones serve as substrate scanners that bring the substrate to Hsp70, simultaneously stimulating the otherwise low basal ATPase activity of Hsp70 [15]. Another co-chaperone, Hsp70-Hsp90 organising protein (Hop) [16,17], is a module that facilitates binding of Hsp70 and Hsp90 to allow Hsp70 to pass substrates to Hsp90. The C-terminal EEVD/N motifs of Hsp70 are thought to be important for its interaction with Hop [18,19,20].

Residues of Hsp70 that are in direct contact with the peptide substrate of Hsp70 are located in the highly conserved β-subdomain of its SBD [21]. The most diverse part of the SBD of the protein is its 10 kDa C-terminal α-helical “lid”. However, the role of the lid of Hsp70 has for a long time been enigmatic until a study established that a lidless form of DnaK (*E. coli* Hsp70) though functional was less effective at supporting lambda growth in vivo [22]. In addition, an independent study demonstrated that a lidless form of DnaK was unable to refold a misfolded protein in vitro [23]. It has further been proposed that some residues located in the α-helical lid subdomain of Hsp70 make ionic contacts with some residues in the β-sheet, suggesting a role for the lid in substrate binding [24,25]. The lid of Hsp70 is further implicated in regulating both substrate selection and the lifespan of the Hsp70-substrate complex [26,27].

One of the defining features of Hsp70 is the presence of GGMP repeat residues (GGMP motif) located in the loop region that is adjacent Helix E of α-helical lid. The role of the GGMP repeats remain largely unknown. Initially, these tetrapeptide repeats were thought to be present only in stress-inducible Hsp70s. However, in some species the GGMP residues are present in both inducible and constitutive forms of Hsp70 [28]. While the GGMP repeats are well represented in cytosolic Hsp70s of parasitic origin including PfHsp70-1, they only occur in a constitutive human Hsp70 homologue [29]. In addition, whereas PfHsp70-1 possesses six GGMP repeats including an additional GGMN segment, the human Hsp70 possesses only two GGMP repeats [29]. Thus, GGMP repeats present PfHsp70-1 and its homologues resident in cytosols of parasitic organisms constitute a distinct feature of these proteins in comparison to the human homologues.

We previously demonstrated that PfHsp70-1 possesses chaperone activity and that it physically associates with *P. falciparum* Hsp90 via Hop (PfHop) as a module [17,30]. In addition, we recently established that PfHsp70-1 exhibits unique structure-function features compared to DnaK [13]. Whereas GGMP repeats are present in PfHsp70-1, they are missing in DnaK [29]. One of the unique functional features of PfHsp70-1 is that it binds to asparagine enriched peptide substrates [13]. Interestingly, it is estimated that 30% of *P. falciparum* proteome is glutamate/asparagine rich and hence prone to misfolding and aggregation [31,32]. We therefore hypothesized that the GGMP motif may play an important role in regulating the chaperone function of parasite Hsp70s.

In the current study, we introduced GGMP mutations in both PfHsp70-1 and its derivative chimeric protein, KPf (comprised of the ATPase domain of DnaK fused to the SBD of PfHsp70-1) [30]. Introduction of the GGMP mutations in KPf, abrogated its capability to protect *E. coli dnaK756* cells [33] cultured at elevated growth temperature. On the other hand, insertion of the GGMP motif in *E. coli* DnaK led to a lethal phenotype under heat stress growth conditions. We further conducted a battery of biochemical and biophysical assays and established that GGMP repeats are important not only for substrate binding but also play an important role in facilitating direct interaction of PfHsp70-1 and its co-chaperone, PfHop. We discuss the significance of the GGMP motif in regulating functional specificity of PfHsp70-1.

## 2. Results

### 2.1. GGMP Repeat Motifs Are Predominantly Associated with Hsp70s of Apicomplexan Species

Sequence alignment data showed that cytosol localised Hsp70s of members of the phylum, Apicomplexa, are endowed with the most prominent GGMP repeats. Notably, human Hsc70 possesses only two GGMP repeats compared to several of these repeat motifs present in Hsp70s of *Cryptosporidium parvum* (eleven), *Trypanosoma cruzi* (ten), *Plasmodium berghei* (nine), and *Toxoplasma gondii* (six), respectively (Figure 1A). In addition, there is a strong presence of GGMP motifs in all *Plasmodium* species, with frequencies of the repeats ranging from five to nine across the various members of these genus (Figure 1A). *P. berghei* and *P. vivax* Hsp70s harbour the highest number of GGMP repeats. PfHsp70-1 possesses seven GGMP repeats and of these four are strictly conserved GGMP segments while the rest are interrupted with occasional substitute residues. While the role of the GGMP motif remains unclear, a previous study suggested that the GGAP motif of yeast Hsp70 may regulate its interaction with substrate and co-chaperones [34]. Interestingly, while Hsp70 from *Saccharomyces cerevisiae* is characterised by three GGAP repeat motifs, members of Apicomplexa are endowed with cytosolic Hsp70s that habour pronounced GGMP motifs (Figure 1B). Therefore, it appears that the GGMP motif of Hsp70s of apicomplexans may play a more prominent and distinct role. Further highlighting the unique presence of these motifs in apicomplexans, we noted that GGMP repeats are absent in Hsp70s of other human pathogens such as bacteria (Figure 1A).

### 2.2. Three-Dimensional Models of Wild Type Hsp70 versus GGMP Variants

The three-dimensional models of wild type PfHsp70-1 and its GGMP variants (Figure 1B) were analysed to predict structural variations that could have been introduced by the GGMP mutations (Figure 2). Furthermore, we also mapped out unique hydrogen bonding patterns present in the GGMP mutants in comparison to the wild type protein (Appendix A). In comparison to that of wild type protein, the three-dimensional model of PfHsp70-1_G632_ revealed slight variations in the spatial orientation of the EEVD motifs and the loops adjoining the SBDβ (Figure 2A). However, there were no variations in the hydrogen bonding involved in stabilizing the SBDβ segments of the two proteins (Appendix A). At least 6 hydrogen bonds unique to the SBDα of PfHsp70-1_G632_ relative to the wild type protein were observed (Appendix A). These H bonds unique to the PfHsp70-1_G632_ variant appear to have modulated the orientation of the C-terminal EEVD motif (Figure 2A).

Minor spatial alterations were also observed in the SDBβ loop region of PfHsp70-1_G648_ relative to the wild type protein (Figure 2B). However, an additional helical segment appears to have been introduced within the SBDα of PfHsp70-1_G648_ involving residues ^647^PAALD^651^ (Figure 2B zoomed panel). This helical section, absent in PfHsp70-1, was predicted to be stabilized by 8 H bonds (Appendix A). In addition, the EEVD motif of the PfHsp70-1_G648_ variant subsequently underwent a positional reorientation relative to the wild type form (Figure 2B). This EEVD motif re-orientation could possibly affect crosstalk of the mutant with co-chaperones.

PfHsp70-1 and PfHsp70-1_G632-664_ exhibited marked variations in their SBDβ segments (Figure 2C). Notably, these variations are characterised by unique H-bonding patterns originally absent in wild type PfHsp70-1 but present in the mutant protein. PfHsp70-1_G632-664_ exhibited unique loop-reorientations at the L_1,2_, L_3,4_ and L_5,6_ possibly leading to varied peptide substrate binding properties of PfHsp70-1_G632-664_ mutant relative to the wild type protein (Figure 2C; Appendix A). Minor adjustments in the orientation of loops located in the substrate binding cavity of Hsp70 have been shown to alter substrate binding affinity [35,36,37]. PfHsp70-1_G632-664_ possesses a unique H bond between N atom of T^428^ and O atom of L^468^ (with distance 3.385 Å) which is absent in the SBDβ of the wild type form. This bond connects loops L_3,4_ and L_5,6_ (Figure 2C) possibly enhancing the stability of the substrate binding cleft of PfHsp70-1_G632-664_. PfHsp70-1G_632-664_ also possesses a unique T^428^-A^430^ (3.253 Å) H-bond in L_1,2_ that occurs between the two oxygen atoms. We also observed that L_5,6_ of PfHsp70-1_G632-664_ forms a unique H-bond with L_3,4_. The predicted structural variations between the SBDβ subdomains of wild type PfHsp70-1 relative to PfHsp70-1G_632-664_ would possibly lead to variable peptide binding affinities. Notably, in comparison to PfHsp70-1G_632-664_, wild type PfHsp70-1 exhibits a slightly longer loop extending towards the C-terminus which harbours the GGMP motifs. H-bonding analysis of helix E revealed that PfHsp70-1_G632-664_ gained 2 extra H-bonds between residues P^562^-E^566^ (2.932 Å) and Q^561^-I^565^ (2.617 Å) (Appendix A) in comparison to the wild type protein. The presence of these bonds could slightly enhance the stability of the lid section of this protein. Additionally, the C-terminal loop of PfHsp70-1_G632-664_ had an extra H-bond formed by residues I^627^-A^650^, resulting in a distinct spatial re-orientation of the C-terminal EEVD motif.

Comparisons of the three-dimensional models of PfHsp70-1 and PfHsp70-1_ΔG_ revealed variations in the SBDβ loop segments of the proteins (Figure 2D). Expected of such a drastic mutation, PfHsp70-1_ΔG_ lacked several H-bonds in L_1,2_ present in the wild type protein (Appendix A). For example, two H-bonds that are predicted to form between β3 and β5 of wild type PfHsp70-1 were abrogated in PfHsp70-1_ΔG_ (Appendix A). This suggests that deletion of the GGMP motif from PfHsp70-1 possibly reduced the stability of the SBDβ segment of the protein, potentially impacting on this mutant’s capacity to bind peptide substrates. Notably, PfHsp70-1_ΔG_ also lost the helical structure of its lid segment extending to the C-terminus (Figure 2D). Additionally, although the deleted GGMP segment was positioned between residues 632 and 664, it should be noted that all the H-bonds that stabilize helix E of the lid starting from residue E^535^ were predictably disrupted (Appendix A). As a result, the C-terminal EEVD motif of PfHsp70-1_ΔG_ was repositioned. As such the deletion mutation could impact on co-chaperone binding.

We recently established that PfHsp70-1 and *E. coli* DnaK exhibit unique structure-function features [13]. Although the SBD is known to play a role in defining the functions of these two proteins, the role of the GGMP in this regard is unknown. As such, we wanted to enquire how insertion of the GGMP repeats of PfHsp70-1 into DnaK would impact on the structure-function features of the latter. Comparative analyses of the three-dimensional models of DnaK versus its GGMP insertion mutant, DnaK_G_, showed that there were differences within the loops of SBDβ (Figure 2E). However, the changes did not appear to influence H bonding within the SBDβ section of DnaK. Notably, insertion of the GGMP segment into DnaK induced loss of the α-helical structure from position ^617^QQQH-QQTA^627^ which constitutes helix E of SBDα (Figure 2E). As such, DnaK_G_ possesses a longer C-terminal loop marked by residues ^617^QQQH-KDKK^673^ (Figure 2E). Thus, the inserted GGMP while not disrupting the SBDβ, appears to affect the conformational integrity of the lid segment of DnaK.

### 2.3. GGMP Repeat Motifs Are Important for the Conformational Stability of PfHsp70-1

Following the successful purification of the wild type proteins and their derivatives (Figure 1B and Appendix A), we sought to establish the effect of the GGMP mutations on the conformations of the proteins by circular dichroism (CD) spectroscopic analyses. We further conducted the same assays on the DnaK-_G_ derivative relative to wild type DnaK. Overall, the GGMP mutants of PfHsp70-1 exhibited similar CD spectra to that of wild type PfHsp70-1 protein (Figure 3A) characterised with minima at 222 nm and 209 nm, and maxima at 194 consistent with the reported α-helical nature of the protein [13]. This suggests that the wild type as well as the two select GGMP derivatives of PfHsp70-1 proteins were properly folded. Consistent with previous reports [38], we observed that PfHsp70-1 is predominantly α-helical (62%). The variant PfHsp70-1_G632_ exhibited a markedly reduced 54% α-helical content than the wild type protein. In addition, the mutants PfHsp70-1_G632-664_ and PfHsp70-1_ΔG_ exhibited 55% and 52% α-helical content respectively, which was lower than that of wild type protein. On the other hand, PfHsp70-1_G648_ exhibited a higher (69%) α-helical content than the wild type protein which is consistent with the in-silico data which showed that this mutant gained an additional helical segment involving residues, ^647^PAALD^651^ (Figure 2B).

PfHsp70-1 is known to be fairly heat stable [13]. Similarly, its two GGMP variants PfHsp70-1_G648_, PfHsp70-1_ΔG_, exhibited stability to heat stress comparable to that of wild type protein monitored at both 222 nm (Figure 3B). As previously reported [13,38], PfHsp70-1 displays two melting transitions when monitored at 222 nm. The first phase is gradual and occurs at around 40–55 °C, while the second step is steeper and occurs at 56 °C to 65 °C (Figure 3B). The first transition is known to represent unfolding of the less heat stable ATPase domain, while the second phase represents the unfolding of the SBD [38]. Interestingly, derivatives, PfHsp70-1_G632_, PfHsp70-1_G648_, and PfHsp70-1_G632-664_ exhibited a more drastic first transition step (40 °C to 55 °C) which was slightly steeper than that of wild type protein. On the other hand, the second transition step of PfHsp70-1_G632_ and PfHsp70-1_G632-664_ covering temperature range 58 °C to 90 °C was narrower than that of the wild type protein. The second transition step of PfHsp70-1_G648_ was like that of wild type. This suggests that GGMP changes introduced in PfHsp70-1_G632_ and PfHsp70-1_G632-664_ resulted in a less stable ATPase domain while at the same time the changes slightly reduced its SBD stability. Similarly, PfHsp70-1_G648_ exhibited a less stable ATPase domain compared to wild type protein. However, its second melting profile representing the unfolding of the SBD resembled that of the wild type protein. The GGMP deletion derivative, PfHsp70-1_ΔG_, exhibited a similar first melting curve compared to the wild type while its second melting curve was broader than that of wild type. This suggests that deletion of GGMP repeat residues did not result in loss of its conformational stability.

We further explored the effect of inserting the GGMP repeats of PfHsp70-1 into DnaK to generate, the mutant, DnaK_-G_. While wild type DnaK displayed a fold profile like one we observed before [13] (Figure 3C), the GGMP insertion derivative exhibited a shallower CD spectrum at 209 and 222 nm. This shows that the mutant had less α-helical content (55%) than that of the wild type protein which registered α-helical content of 59%. Notably, DnaK-_G_ exhibited a steeper first melting curve at temperatures up to 45 °C and a more rapid curve up to 65 °C (Figure 3D). This suggest that insertion of the GGMP repeats reduced the stabilities of both the SBD and ATPase domains of DnaK.

### 2.4. The GGMP Repeats Contribute towards Basal ATPase Activity of PfHsp70-1

The basal ATPase activities of the proteins were determined by measuring the amount of inorganic phosphate (Pi) released. The kinetics determined from Michaelis-Menten curves revealed that PfHsp70-1 possesses higher basal ATP turnover rate as compared to the variants (Figure 4A, Appendix A, *p* < 0.001). The findings suggest that the GGMP repeats contribute to the ATPase activity of PfHsp70-1. However, PfHsp70-1_G648_, was slightly more active than the rest of the GGMP mutants of PfHsp70-1 (*p* < 0.05). This suggests that the input of the GGMP repeat segment on the ATPase activity of the protein depends on the position that the motif occupies. PfHsp70-1 is known to possess much higher ATPase activity than that of its human homologue [39,40]. The pronounced presence of GGMP repeats in PfHsp70-1 compared to human Hsp70 may account for this phenomenon. To support this further, introduction of the GGMP motif of PfHsp70-1 in DnaK resulted in its mutant, DnaK-_G_, exhibiting higher basal ATPase activity than the wild type protein (Figure 4A).

### 2.5. The GGMP Repeat Segment of PfHsp70-1 Is Essential for Chaperone Function

As holdase chaperones, PfHsp70-1 and DnaK are known to suppress heat-induced aggregation of model substrate proteins [3,41,42]. We therefore explored the effect of the GGMP mutations on the capability of the chaperones to suppress heat induced aggregation of an aggregation prone protein, MDH, in vitro. As expected, both PfHsp70-1 and DnaK suppressed the aggregation of MDH in the absence of nucleotide (NN) and in the presence of ADP (Figure 4B). Furthermore, as expected ATP inhibited the chaperone (suppression of protein aggregation) of both PfHsp70-1 and DnaK.

As observed for their wild type forms, all the GGMP mutants of PfHsp70-1 and DnaK had their holdase chaperone activity inhibited by ATP (Figure 4B). This demonstrates that these mutants retained a level of allosteric function. PfHsp70-1_G632_ and PfHsp70-1_G648_ were both capable of suppressing MDH aggregation in the *apo* and ADP states. However, as holdase chaperones, the GGMP derivatives, PfHsp70-1_G632-664_ and PfHsp70-1_ΔG_, were functionally less effective than wild type PfHsp70-1 (*p* < 0.01; Figure 4B). While DnaK-_G_ exhibited less holdase chaperone activity than wild type DnaK in the *apo* state (*p* < 0.01), this drop in activity was not statistically significant in the presence of ADP (*p* > 0.05; Figure 4B).

To further confirm the role of the GGMP repeat segment of PfHsp70-1, we investigated the capacity of GGMP mutants of its chimera, KPf, to protect *E. coli dnaK756* cells subjected to heat stress at 43.5 °C. First, we confirmed that KPf and DnaK both heterologously expressed in *E. coli dnaK756* cells were capable of supporting the growth of these cells at 43.5 °C. As expected, a negative control, constituted by the cells heterologously expressing the mutant V436F with a hydrophobic pocket substitution led to cell death [43]. Cells expressing the derivatives, KPf_G617_ and KPf_G633_ (that mirrored mutants PfHsp70-1_G632_ and PfHsp70-1_G648_, respectively) did grow at 43.5 °C (Figure 4C). This suggests that these mutants demonstrated chaperone function within a cellular setting. This was in line with in vitro generated data that demonstrated that mutants, PfHsp70-1_G632_ and PfHsp70-1_G648_ retained their holdase chaperone activity. However, the extended GGMP substitution and deletion mutants, KPf_G617-649_ and KPf_ΔG_ failed to confer cyto-protection to *E. coli dnaK756* cells at 43.5 °C. In line with this, PfHsp70-1 mutants carrying the same mutants exhibited slightly compromised in vitro chaperone activity. To our surprise, whereas the mutant, DnaK-_G_, exhibited in vitro chaperone activity comparable to that of wild type protein (Figure 4B), it was not able to confer cyto-protection to *E. coli dnaK756* cells at 43.5 °C. This demonstrates that although DnaK-_G_ was passively able to inhibit the heat-induced aggregation of MDH in vitro, it was ineffective at conferring cyto-protection to *E. coli* cells grown at the elevated temperature of 43.5 °C. However, the chaperone function exhibited by DnaK-_G_ in vitro showed that it was expressed in *E. coli* in folded functional form. That *E. coli dnaK756* cultured at permissive temperature (37 °C) and expressing DnaK-_G_ thrived at this temperature, demonstrated that this protein was produced in non-toxic form. The findings thus suggest that the GGMP motif of PfHsp70-1 compromised the function of *E. coli* DnaK, abrogating its capability to support growth at elevated temperature. Altogether, the findings show that GGMP repeat segments work in concert to support the chaperone function of PfHsp70-1. In addition, we established an essential role of this motif for the chaperone function of the SBD of this protein in *E. coli dnaK756* cells. However, we also established that although DnaK carrying the GGMP repeat segment of PfHsp70-1 exhibits function in vitro, it is functionally compromised in cells cultured at elevated temperatures. This suggests that the GGMP repeat segment is an important functional motif that specifically supports chaperone function of Hsp70s of parasitic organisms such as members of the Apicomplexa phylum. However, DnaK does not need this motif to carry out its function. We previously observed that PfHsp70-1 and DnaK are functionally distinct [13]. We speculate based on the current findings that the GGMP motif is a possible determinant of PfHsp70-1 function.

### 2.6. The GGMP Repeat Motif Is Important for Substrate Binding

We have previously demonstrated that Hsp70s of plasmodial origin preferentially bind to peptides that are asparagine rich [13,20]. In the current study, we investigated the effect of GGMP mutations on the capability of PfHsp70-1 to bind to a model peptide, ANNNMYRR [13,20], using SPR analysis (Figure 5A–C; Appendix A).

PfHsp70-1 bound to the peptide substrate within micromolar range of affinity (Appendix A). Nucleotide dependent interaction was observed as there was a single order of magnitude drop in affinity in the presence of ATP as expected [13]. In the *apo* state, both PfHsp70-1_G632_ and PfHsp70-1_G648_ exhibited comparable substrate affinity to wild type protein. However, in the presence of ADP, PfHsp70-1_G632_ registered a drop in affinity in comparison to both wild type protein and the mutant, PfHsp70-1_G648_ (Figure 5D). Both the extended GGMP substitution (PfHsp70-1_G632-664_) and GGMP deletion (PfHsp70-1_ΔG_) mutants, drastically lost affinity for the peptide substrate relative to wild type protein and the other two mutants (*p* < 0.001) (Figure 5; Appendix A). Notably, the affinity of PfHsp70-1_G632-664_ and PfHsp70-1_ΔG_ for peptide was comparable in *apo*, ADP and ATP states. This suggests that the extended GGMP mutation and deletion mutations led to non-specific peptide binding. It is possible that the changes compromised the allosteric function of the protein. The degree of the functional compromise of these two mutants is corroborated by the loss of chaperone function exhibited by similar mutations introduced in KPf as demonstrated by the complementation assay (Figure 4C). Overall, these findings suggest that the GGMP repeat segments are important for substrate binding.

### 2.7. GGMP Mutations Compromise Interaction of PfHsp70-1 with PfHop

The functional interaction of PfHop with PfHsp70-1 was previously demonstrated based on pull down and direct in vitro SPR based assays [17,19]. Since the GGMP repeat segment is positioned in proximity to the C-terminal EEVD motif, we surmised that GGMP mutations may impact on interaction of PfHsp70-1 with its co-chaperone, PfHop. The C-terminal EEVD motif is known to be crucial for Hsp70-Hop interaction [20]. SPR measurements showed that PfHsp70-1 interacted with PfHop at nanomolar to upper micromolar ranges in the *apo*/ADP states as previously reported (Figure 6; Appendix A) [19].

In the presence of ATP, the affinity of PfHsp70-1 for PfHop was reduced to levels within the lower micromolar range compared to the *apo* or ADP states (Appendix A), consistent with a previous study which established that PfHop affinity for PfHsp70-1 is higher in the *apo*/ADP state than in the ATP state [19]. Following this, comparative binding affinities of the GGMP variants relative to PfHsp70-1 for PfHop were determined (Figure 6D). In the *apo* and ADP states, all the GGMP mutants registered a drop in affinity for PfHop relative to the wild type PfHsp70-1. Notably, PfHsp70-1_G632-664_ and PfHsp70-_∆G_ registered the biggest losses in affinity for PfHop (Figure 6D, Appendix A). Interestingly, under all conditions (*apo*, ADP, ATP), the extended GGMP substitution mutant, PfHsp70-1_G632-664_, interacted with PfHop within micromolar range of affinity (Appendix A). Thus, PfHop interaction with PfHsp70-1_G632-664_ was not regulated by nucleotide. This suggests that this mutant may have a compromised allosteric function. The affinity of GGMP deletion mutant, PfHsp70-_∆G_, for PfHop was within the micromolar range in the *apo* and ATP states (Appendix A). Altogether, this suggests that both the extended GGMP substitution, PfHsp70-1_G632-664_, and deletion mutant, PfHsp70-_∆G_, interacted with PfHop in atypical fashion. This implies that these mutations compromised the interaction of these two mutants with PfHop.

### 2.8. The GGMP Motif of PfHsp70-1 Is Insufficient to Promote the Association of PfHop with DnaK

Both ELISA and slot blot based data show that there was no interaction between PfHop with either DnaK or DnaK-_G_ (Figure 7A–C). There were no statistically significant differences between the control (BSA-PfHop) compared to DnaK-PfHop/DnaK-_G_-PfHop signals (Figure 7D). The slot blot data further validated lack of interaction between PfHop and DnaK/DnaK-_G_ (Figure 7D). Densitometric analyses of band intensities were used to validate the slot blot data (Figure 7D). This suggests that although the GGMP repeat residues of PfHsp70-1 are important for its interaction with PfHop, they are insufficient to facilitate this association. This implies that other segments of PfHsp70-1 such as C-terminal EEVD motif are important for this interaction. DnaK possesses EEVKDK residues in place of the EEVD residues present in cytosolic Hsp70s of eukaryotes. It is possible that the presence of the residues K and D, in the C-terminal fragment of DnaK, does not promote this association. Since *E. coli* does not possess a Hop homologue, DnaK does not need residues for interaction with Hop. Thus, by inserting GGMP repeat residues of PfHsp70-1 into DnaK, we established that other segments of PfHsp70-1 are required to cooperate with GGMP residues to bind PfHop.

## 3. Discussion

The architecture of Hsp70 is marked by two distinct domains: N-terminal NBD/ATPase domain and the C-terminal SBD. Hsp70s of parasitic origin, particularly members of the Apicomplexan family are characterised by the presence of GGMP repeat segments that are embedded within their C-terminal SBD [29,39]. The role of the GGMP motif in regulating the chaperone function of these Hsp70s remains unknown. In the current study, we established that GGMP repeats are present in Hsp70s of parasites and are particularly enhanced in species belonging to the Apicomplexa phylum. Based on bioinformatics, biophysical, biochemical and complementation assays, we established for the first time that the GGMP motif of PfHsp70-1 is important for the ATPase activity of the protein and regulates other key functional aspects such as substrate binding, chaperone function and the interaction of the protein with its co-chaperone, PfHop. Notably, the GGMP motif is absent in other pathogenic organisms, especially bacteria. In addition, only one human Hsp70 (Hsc70) contains two GGMP repeats compared to *P. falciparum* Hsp70 which possesses about seven of these repeats. For this reason, we speculate that the GGMP motif of *P. falciparum* may account for reported specialised functional features of this protein [13,39]. In *Toxoplasma gondii*, there were more GGMP repeats in Hsp70 from a more virulent strain compared to a less virulent one, leading to speculation that this motif may regulate pathogenicity [44].

PfHsp70-1 exhibits key functional features (Figure 8) such as notably high basal ATPase activity and is generally more efficient as a chaperone than its human and *E. coli* counterparts [13,39,40]. It also possesses high stability to heat stress and binds to N-enriched peptides with higher affinity than *E. coli* DnaK [13]. In the current study, we observed that mutation or deletion of the GGMP repeats appears to lead to two key structural defects; reorientation of the SBDβ loops and spatial re-orientation of the lid and the C-terminal EEVD motif (Figure 2 and Figure 8). The reorientation of the SBDβ loops as well as that of the lid and the C-terminal EEVD motif reduced affinity for substrate, consequently, compromising the otherwise efficient chaperone function of the protein. In addition, the spatial reorientation of the lid and the EEVD motif may account for various functional defects of the protein such as lower conformational stability to stress, reduced affinity for the co-chaperone, PfHop, and the lower ATPase activity it registered. Although the GGMP mutants retained some degree of allosteric function based on suppression of MDH aggregation (Figure 4), data on their interaction with PfHop suggested that PfHsp70-1_G632-664_ and PfHsp70-_∆G_ may have lost some degree of allosteric function (Figure 6). We thus speculate that deletion/mutation of GGMP residues impacted more adversely on lid-mediated allostery while the protein retained some degree of linker mediated allostery. The loss of lid-mediated allostery may account for the observed reduction in both ATPase activity and thermal stability registered by some of the GGMP mutant forms of the protein, as reviewed in [45]. In support of this, we previously demonstrated that the C-terminal EEVN motif of another plasmodial Hsp70, PfHsp70-x, was important for both stability and ATPase function of the protein [20]. Notably, however, the extended GGMP substitution mutant was less stable than GGMP deletion mutant (Figure 3). The stability of the deletion mutant could be accounted for by the more compact nature of its SBD as well as the effect of the role of hydrogen bonding between the repositioned lid residues and those of the SBDβ subdomain.

Using recombinant forms of the proteins, we conducted CD analyses (Figure 3). Generally, CD spectra suggested that the mutations did not drastically affect the conformations of the protein. However, it appears that the GGMP repeat motifs may influence the stability of the protein to heat stress. This is important as Hsp70 acts as a thermosensor for cellular stress response [46]. This role is particularly important in malaria parasites as they experience cellular stress associated with the febrile fever episodes associated with clinical malaria progression [47]. Thus, the possible role of the GGMP motif in conferring heat stability to PfHsp70-1 is important for its cytoprotective function. Interestingly, an independent study proposed that the C-terminal SBD of PfHsp70-1 which harbours the GGMP motif is responsible for the heat stability of the protein [38].

A recent study proposed that PfHsp70-1 is a more efficient refolding chaperone than human Hsp70 [40]. It is thus possible that the presence of the GGMP repeats in PfHsp70-1 and other cytosol-localized Hsp70s of parasitic origin that are endowed with this motif may augment the functional efficiency of these chaperones, to ensure survival in the host as well as regulate virulence. In addition, the presence of the GGMP motif in PfHsp70-1 could tailor the chaperone to meet the specialized protein folding demands of the parasite proteome exacerbated by the dominant presence of Q and N residues [13,31,32].

With respect to the in vitro holdase chaperone assay, both the extended GGMP substitution and deletion mutants of PfHsp70-1 were less effective at suppressing MDH aggregation. To back this up, *E. coli dnaK756* cells heterologously expressing KPf harboring the extended GGMP substitution and deletion mutations died upon being subjected to heat stress. Interestingly, insertion of the GGMP repeat of PfHsp70-1 into DnaK abrogated its function based on the complementation assay. Since DnaK-_G_ suppressed MDH aggregation in vitro, the lack of function of this protein in a cellular setting suggests that the GGMP of PfHsp70-1 may have compromised certain functional features of the protein that are not essential for its in vitro holdase function such as interaction with co-chaperones. Residues, ^623^DDVVDAEFEEVKDKK^638^, of DnaK are known to serve as secondary peptide binding site located in the lid segment of the protein [48]. While we inserted the GGMP segment of PfHsp70-1 close to the N-terminal side of this motif, the changes did not disrupt the sequence integrity of this secondary peptide binding site of DnaK. However, it is possible that the presence of the GGMP of PfHsp70-1 in DnaK disrupted the conformational integrity of the lid of DnaK, thus compromising its chaperone function. It has further been suggested that a GGAP motif of yeast Hsp70 could serve as a secondary site for substrate binding [34]. Our current data suggest that the GGMP repeats of PfHsp70-1 are important for substrate binding.

PfHsp70-1 functionally interacts with its co-chaperone, PfHop [17]. Since this interaction is primarily thought to occur via the C-terminal EEVD motif of PfHsp70-1, we investigated the effects of the GGMP mutations on the interaction of PfHsp70-1 with PfHop. We established that PfHsp70-1_G632_ and PfHsp70-1_G648_ mutants exhibited less affinity for PfHop than wild type protein. The extended GGMP substitution mutant (PfHsp70-1_G632-664_) and deletion mutant (PfHsp70-_∆G_) both interacted with PfHop in an atypical fashion that was not regulated by nucleotide. This further validates the importance of the GGMP repeats in regulating Hsp70-Hop interaction.

Furthermore, observations from our work using the DnaK_-G_ variant highlight that although GGMP repeat residues augment interaction of PfHsp70-1 with PfHop other segments of PfHsp70-1 such as the C-terminal EEVD motif are important for this process (Figure 7). Indeed, Hsp70 interaction with Hop is thought to be complex involving several segments of the chaperone [49].

Findings from the current study highlight the essential role of the GGMP repeat segment of PfHsp70-1 in the chaperone function of the protein and in interaction with the co-chaperone, PfHop. In light of the prominent presence of the GGMP motifs in Hsp70s of some parasitic organisms, the findings point to a possible role of this motif in providing bespoke chaperone function in these parasites. Given the unique presence of this motif in some Hsp70s of parasitic organisms, selective targeting of this motif in anti-parasitic drug design may be a viable future prospect.

## 4. Materials and Methods

### 4.1. Determination of the Frequency of GGMP Repeats of Hsp70 across Species

We first sought to establish the frequency of GGMP repeats in Hsp70 proteins of human origin versus those of select parasitic organisms, including members of the Apicomplexan family. FASTA sequences were retrieved from NCBI (https://www.ncbi.nlm.nih.gov/protein/ (accessed on 14 January 2021)) or Uniprot (https://www.uniprot.org (accessed on 14 January 2021)) and multiple sequence alignments were conducted using Jalview version 2 [50].

### 4.2. Comparative Analysis of the Three-Dimensional Models of PfHsp70-1 and DnaK Relative to Their GGMP Variants

Before expressing and purifying recombinant forms of PfHsp70-1 and DnaK as well as their GGMP mutants, we first conducted in silico studies to predict the possible implications of introducing the mutations on the structures and functions of PfHsp70-1 and DnaK.

The three-dimensional models of wild type PfHsp70-1 and DnaK relative to those of their GGMP mutant versions were generated using PHYRE^2^ (http://www.sbg.bio.ic.ac.uk/phyre2 (accessed on 14 January 2021)) [51]. Models of the respective proteins were retrieved from PHYRE^2^ as PDB files and visualised using Chimera version 1.9 [52]. Comparative structural super-positioning of the three-dimensional models of wild type proteins versus their GGMP variants was conducted using the SuperPose tool (http://wishart.biology.ualberta.ca/Superpose/ (accessed on 14 January 2021)) [53]. In SuperPose, the comparative structural similarities were determined using root mean square deviation (RMSD) statistics. The intra protein contact site distances were then measured using Chimera version 1.9 [52].

### 4.3. Design of GGMP Mutants and Generation of Plasmid Constructs

The GGMP motifs occur in PfHsp70-1 between the amino acid positions ^632^GGMPGGMPGGMPGGMPGGMNFPGGMPGAGMPGN^664^. In order to establish the role of the GGMP residues, conservative amino acid substitutions were introduced in the protein sequence at various positions as follows: the first GGMP derivative, PfHsp70-1_G632_ comprised of residues ^632^GGMPGGMP^639^ that were substituted by residues ^632^AALAAALA^639^ (Figure 1B). The second derivative, PfHsp70-1_G648_, comprised of residues ^648^GGMNFPGGMP^657^ that were substituted by residues ^648^AALDFPAALA^657^ (Figure 1B). Similarly, the third variant, PfHsp70-1_G632-664_ represented substitution of the GGMP repeat stretch initially made up of residues ^632^GGMPGGMPGGMPGGMPGGMNFPGGMPGAGMPGN^664^ which were substituted by residues ^632^AALAAALAAALAAALAAALDYAAALAAGALAAD^664^ (Figure 1B). The fourth variant, PfHsp70-1_∆G_, is a deletion mutant of the above described GGMP motif of PfHsp70-1 originally represented by residues 632-664 (Figure 1B). To establish the effects of the GGMP motif on a non-GGMP possessing Hsp70, *E. coli* DnaK, the GGMPGGMPGGMPGGMP motif of PfHsp70-1 was inserted into DnaK to substitute residues ^610^QTAGADA^616^ to generate a mutant, DnaK_-G_ with the inserted residues positions located at the following segment of the resultant protein: ^610^GGMPGGMPGGMPGGMPGGMP^629^ (Figure 1B). *E. coli* codon harmonized forms of DNA segments encoding for the respective proteins were synthesized by GenScript, Piscataway, NJ, USA) and cloned in pQE30 plasmid vector (Qiagen, Frederick, MD, USA) in frame with an N-terminal (His)_6_-tag.

### 4.4. Complementation Assay

We previously used *E. coli dnaK756* cells to conduct complementation assays [30]. While *E. coli dnaK756* cells grow at ambient temperatures they die at elevated temperatures due to compromised DnaK function [33]. In previous studies, we confirmed that a chimeric chaperone, KPf (made up of the ATPase domain of DnaK and the SBD of PfHsp70-1), protected *E. coli dnaK756* cells at elevated temperatures of growth [30,43]. In addition, because KPf possesses DnaK ATPase domain, it interacts with *E. coli* chaperones such as DnaJ [43]. In addition, the presence of the C-terminal SBD of PfHsp70-1 in KPf made it convenient for us to introduce GGMP mutations that mirror those we introduced in the full length PfHsp70-1 protein as described above. The coding sequence of KPf is hosted on pQE60 vector (Qiagen, Frederick, MD, USA) and as such inserts encoding for its GGMP mutants were codon harmonised and synthesized (GenScript, Piscataway, NJ, USA) and cloned onto the pQE60 plasmid vector. Briefly, the following constructs encoded for following KPf GGMP derivatives: pQE60/KPf_G617_ (expressing KPf_G617_ protein, an equivalent of the PfHsp70-1_G632_ mutant), pQE60/KPf_G633_ (expressing KPf_G633_ protein, an equivalent of the PfHsp70-1_G648_ mutant), pQE60/KPf_G617-649_ (expressing KPf_G617-649_ protein, an equivalent of the PfHsp70-1_G632-664_ mutant), pQE60/KPf_ΔG_ (expressing KPf_ΔG_ protein, an equivalent of the PfHsp70-1_ΔG_ mutant). As negative control, we employed a previously described construct, pQE60/KPf_V436F_, expressing KPf possessing a hydrophobic pocket substitution, V436F, which makes this chaperone incapable of cytoprotecting *E. coli dnaK756* cells at elevated temperatures [43]. All the inserts were cloned using *Bam*HI and *Hind*III restriction sites present in the multiple cloning sites of both pQE60 and pQE30.

The various plasmid constructs were used to transform *E. coli dnaK756* cells. The transformed cells were grown in double strength yeast-tryptone (2YT) in the presence of selection antibiotics as previously described [30]. After the overnight incubation at 37 °C, the cells were transferred into fresh broth and grown under similar growth conditions. At OD_600_ = 0.6, protein expression was induced with 1 mM IPTG. The cells were grown until OD_600_ = 2.0. The cultures were standardized to the same cell density. Serial dilutions were made prior to spotting onto agar plates supplemented with the respective antibiotics and 50 μM IPTG. Incubation of the plates proceeded overnight at 37 °C and 43.5 °C. Recombinant expression of KPf and its derivatives in *E. coli dnaK756* was validated by Western blotting using *α*-PfHsp70-1 antibody [3]. The expression of DnaK and its derivative was confirmed by Western blotting analysis using HRP conjugated monoclonal *α*-His antibodies (SigmaAldrich, Darmstadt, Germany).

### 4.5. Expression and Purification of Recombinant Proteins

Since PfHop is a known co-chaperone of PfHsp70-1 [17], we investigated the effects of the GGMP mutations on the interaction of PfHsp70-1 with this co-chaperone. Expression and purification of recombinant forms of proteins were conducted as previously described [17,19], with minor modifications. GGMP derivatives of PfHsp70-1 expressed well in *E. coli* JM109 cells (Thermofisher Scientific, Waltham, MA, USA) and thus were purified from the same cells.

The transformed *E. coli* cells were grown in Terrific broth (TB) supplemented with TB salts (0.17 M KH_2_PO_4_ and 0.72 M K_2_HPO_4_) and 100 μg/mL ampicillin at 37 °C. Expression was induced using 1 mM IPTG. Cells were harvested 5 h post-induction for purification. Pelleted cells were re-suspended in non-native lysis buffer (10 mM Tris– HC1, pH 7.5, 300 mM NaCl, 10 mM imidazole, 1X Sigmafast Protease Inhibitor, 1 mM 2-β-mercaptoethanol and 1 mg/mL lysozyme) containing 8 M urea at 4 °C for 30 min. The cell lysate was sonicated (amplitude 30%; with 10 s ON and 30 s OFF cycles for 5 min) prior to centrifugation at 6000× *g* for 30 min at 4 °C. The proteins were purified by nickel affinity chromatography following a previously described method [19], with minor modifications in wash buffer I (10 mM Tris-HCl, pH 7.5, 300 mM NaCl, 10 mM imidazole, 1 mM 2-β-mecarptoethanol, 1X Sigmafast) with decreasing urea concentrations from an initial 4 M to 2 M urea concentration. Elution was conducted using elution buffer (10 mM Tris-HCl, pH 7.5, 300 mM NaCl, 500 mM imidazole, 1 mM 2-β-mecarptoethanol containing 1X Sigmafast) in the absence of urea. This was followed by extensive dialysis in buffer (10 mM Tris-HCl, pH 7.5, 300 mM NaCl, 1X Sigmafast, 10% glycerol).

### 4.6. Circular Dichroism and Intrinsic Fluorescence Spectroscopic Analyses

Far-UV CD spectroscopy was used to establish the secondary and tertiary structural features of the GGMP mutants of PfHsp70-1 and DnaK relative to their respective wild type forms. CD analysis was carried out using a Chirascan Plus CD spectrometer equipped with a temperature-controlled Peltier (Applied Photophysics, Surrey, UK) following a previous described method [54]. Spectral scans were recorded from 250 to 190 nm for each protein (2 μM) in a 1-mm path-length quartz cuvette (Hellma, Singapore). To resolve the secondary structures the Dichroweb server [http://dichroweb.cryst.bbk.ac.uk (accessed on 14 January 2021)], was used to deconvolute the spectra into α-helix, β-sheet, β- turn and unordered regions [55,56]. Thermal stability was investigated by monitoring the secondary structure of each protein at 222 nm at varying temperatures (from 20 °C to 90 °C). The folded fraction of each respective protein was expressed as a ratio of molar residue ellipticity [θ] deg.cm^2^.dmol^−1^ at each temperature compared with 20 °C.

### 4.7. ATPase Activity Determination

The ATPase activities of the GGMP mutants relative to wild type proteins were determined by monitoring the release of inorganic phosphate upon ATP hydrolysis following a modified approach [4]. Michaelis-Menten plots were generated to determine kinetics for the ATPase activities of the proteins using GraphPad Prism 6.05 (San Diego, CA, USA). The ATPase activities of the mutants were determined relative to the activity of wild type PfHsp70-1 as reference control.

### 4.8. Assessment of the Effect of GGMP Mutations on the Protein Aggregation Suppression Activities of PfHsp70-1 and DnaK

Hsp70 is capable of suppressing heat-induced aggregation of malate dehydrogenase (MDH) in vitro [3]. We thus investigated the chaperone activities of the GGMP mutant proteins versus that of wild type forms of PfHsp70-1 and DnaK by assessing their capability to suppress aggregation of heat-denatured malate dehydrogenase [MDH (Sigma-Aldrich, Darmstadt, Germany)] as previously described [41,54]. First, it was important to validate the heat stability of the chaperones in the assay buffer and all the chaperone proteins were stable at 51 °C. However, under the same conditions, the substrate (MDH) aggregated spontaneously without chaperone at 51 °C and this reading was set as 100% aggregation. We mixed 0.2 μM of the chaperone. MDH was introduced into the reaction mix to achieve equimolar chaperone and MDH in 1:1 ratio as previously established [54]. Aggregation of MDH was monitored at 51 °C with readings taken every 5 min for 1 h. Absorbance was determined using a SpectraMax M3 spectrometer (Molecular Devices, San Jose, CA, USA). The assay was repeated in the presence of nucleotides (5 mM ATP/ADP).

### 4.9. Determination of Substrate Binding Capabilities

Furthermore, peptide substrate binding affinities of wild type relative to its GGMP derivatives were determined by SPR analysis using a MP-SPR Navi™ 420A ILVES (BioNavis, Finland). A previously described PfHsp70-1 peptide substrate, ANNNMYRR was injected as analyte at varying concentration (0–20 nM) at a flow rate of 50 µL/min as previously described [20]. Ligand-analyte association and dissociation were allowed for 180 s and 300 s, respectively. Steady state equilibrium constant data were determined from the generated sensograms using TraceDrawer (version 1.8). The assays were conducted in the absence or presence of 5 mM ATP/ADP.

### 4.10. Determination of the Effects of the GGMP Mutations on the Interaction of PfHsp70-1 with PfHop

PfHsp70-1 is known to interact with a co-chaperone, PfHop, in a nucleotide dependent manner [19]. SPR analysis was conducted following a previously described protocol [19], to assess the effects of the GGMP mutations on the interaction of PfHsp70-1 and PfHop. Either PfHsp70-1 or its derivatives served as ligands. As analyte, PfHop at varying concentrations (0–1000 nM) was injected at 50 µL/min. Association was allowed to continue for 180 s and dissociation occurred for 300 s. To determine the effect of nucleotides on the interaction of the GGMP variants with PfHop, the assay was repeated in the presence of 5 mM nucleotide (ATP and ADP) as previously described [19].

### 4.11. Investigating the Effect of Inserting GGMP Repeat Residues into DnaK on Its Potential to with PfHop

In a separate experiment, we sought to ascertain the effect of inserting the GGMP motif of PfHsp70-1 into DnaK with respect to interaction of the mutant (DnaK-_G_) with PfHop. We used the GGMP deficient native *E. coli* DnaK and its GGMP variant (DnaK_G_) to conduct slot blot assay and ELISA as previously described [20], towards exploring if the presence of the GGMP repeat residues would facilitate DnaK binding to PfHop. Briefly, the ELISA was conducted by immobilizing 5 μg/mL DnaK and DnaK-_G_ as ligand onto 96 well plates. PfHsp70-1 and BSA were immobilized as positive and negative controls, respectively. Serial dilutions of the analyte recombinant PfHop protein (0–0.5 μg) were prepared in binding buffer (25 mM Tris, 140 mM NaCl, 3.0 mM KCl, 0.1% Tween-20 and 0.1% BSA). The analytes were incubated with the ligand Hsp70 proteins for 2 hr followed by washing steps to remove unbound analyte. The amount of bound analyte was quantified using α-PfHop antibody as primary antibody (1:2000) [19], and an HRP conjugated goat raised α-rabbit IgG was used as secondary antibody (1:4000). Signal development was conducted by adding 3,3′,5,5′-tetramethylbenzidine (TMB) substrate into each well and the reaction was incubated for 2 min with 3-s shaking. Colour development was monitored by measuring absorbance readings at 650 nm using a SpectraMax M3 microplate reader (Molecular Devices, San Jose, CA, USA) at time intervals (0, 5, 10, 15, 20, 25, and 30 min). The absorbance reading for each well was plotted against time. Assays were also repeated in the presence of 5 mM ATP or ADP.

In the Slot blot assay, varying concentrations (1 μg, 2 μg, 4 μg) of recombinant DnaK and its GGMP variant were prepared in buffer T (25 mM Tris-HCl pH 7.4, 140 mM NaCl, and 3.0 mM KCl). Similar controls were used as in the ELISA. The proteins were immobilized onto a nitrocellulose membrane by applying gentle vacuum. The membrane was blocked using 5% non-fat milk made in buffer T plus 0.1% Tween-20. The membrane was then overlaid with 4 μg of purified PfHop protein overnight at 4 °C. All wash steps were performed using buffer T plus 0.1% Tween-20. In order to validate the effect of nucleotides, the assay was repeated in the presence of 5 mM ATP/ADP supplemented in all reaction mixtures. Immunoblotting was conducted using α-PfHop antibody as primary antibody (1:2000) [19], to detect the PfHop and an HRP conjugated goat raised α-rabbit IgG was used as secondary antibody (1:4000). Detection of protein bands was performed using ECL and visualization was done using the Chemidoc (Bio-Rad, Hercules, CA, USA).

### 4.12. Statistical Analysis

The data generated from the assays were analyzed using GraphPad Prism 6.05 software. The statistical analysis was conducted using either one way or two-way ANOVA to determine the statistical significance at (* *p* < 0.05; ** *p* < 0.01; *** *p* < 0.001).

## Figures and Tables

**Figure 1 ijms-22-02226-f001:**
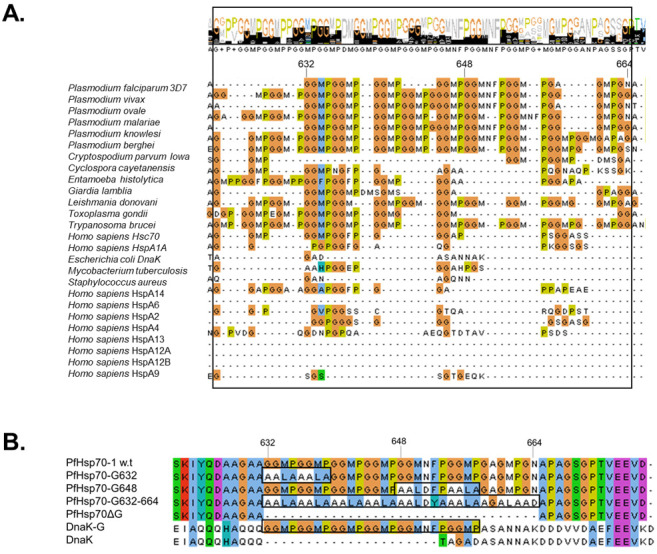
Multiple sequence alignment (MSA) of Hsp70s and design of GGMP mutants. (**A**) MSA of *Plasmodium falciparum* 3D7 Hsp70 (PfHsp70-1, NCBI accession number: XP_001349336.1); *Plasmodium vivax* (NCBI: XP_001614972.1); *Plasmodium ovale curtisi* (GenBank: SBS81157.1); *Plasmodium malariae* (NCBI: XP_028860418.1); *Plasmodium knowlesi* strain H (NCBI: XP_002258136.1); *Plasmodium berghei* ANKA Hsp70 (NCBI: XP_022712526.1); *Cryptosporidium parvum Iowa* II (NCBI: XP_625373.1); *Cyclospora cayetanensis* (NCBI: XP_022588287.1); *Entamoeba histolytica* (GenBank: AAA29102.1); *Giardia lamblia* ATCC 50,803 (GenBank: EDO80296.1); *Leishmania donovani* (UniprotKB: P17804); *Toxoplasma gondii* Hsp70 (UniprotKB: A0A125YXI9); *Trypanosoma brucei* HSP74 (UniprotKB: P11145); *Homo sapiens* heat shock cognate 71 kDa (Hsc70, NCBI: NP_006588.1); *Homo sapiens* Hsp70 protein 1A (HspA1A, NCBI: NP_005336.3); *E. coli* Hsp70 (DnaK, UniProtKB: A1A766.1); *Mycobacterium tuberculosis* Hsp70 (UniProtKB: P9WMJ9.1); *Staphylococcus aureus* HSP70 (GenBank: BAA06359.1); *Saccharomyces cerevisiae* HSP70 (Ssa1p, GenBank: AAC04952.1); *Homo sapiens* heat shock protein 70s (HspA14; NCBI: NP_057383.2), HspA6 (NCBI: NP_002146.2), HspA2 (NCBI: NP_068814.2), HspA4 (NCBI: NP_002145.3), HspA13 (NCBI: NP_008879.3), HspA12A (NCBI: NP_001317093.1), HspA12B (NP_001317093.2), HspA9 (NCBI, NP_004125.3). The black rectangle marks the GGMP repeat segment and the top panel highlights the overall residue conservation level. Residue numbering is based on PfHsp70-1. (**B**) Mutations were introduced at various positions within the GGMP repeat segment of PfHsp70-1 to generate the following derivatives: PfHsp70-1_G632_, PfHsp70-1_G648_, PfHsp70-_1G632-664_, PfHsp70-1_∆G_. The GGMP segment of PfHsp70-1 was inserted into DnaK to generate the derivative, “DnaK-_G_”.

**Figure 2 ijms-22-02226-f002:**
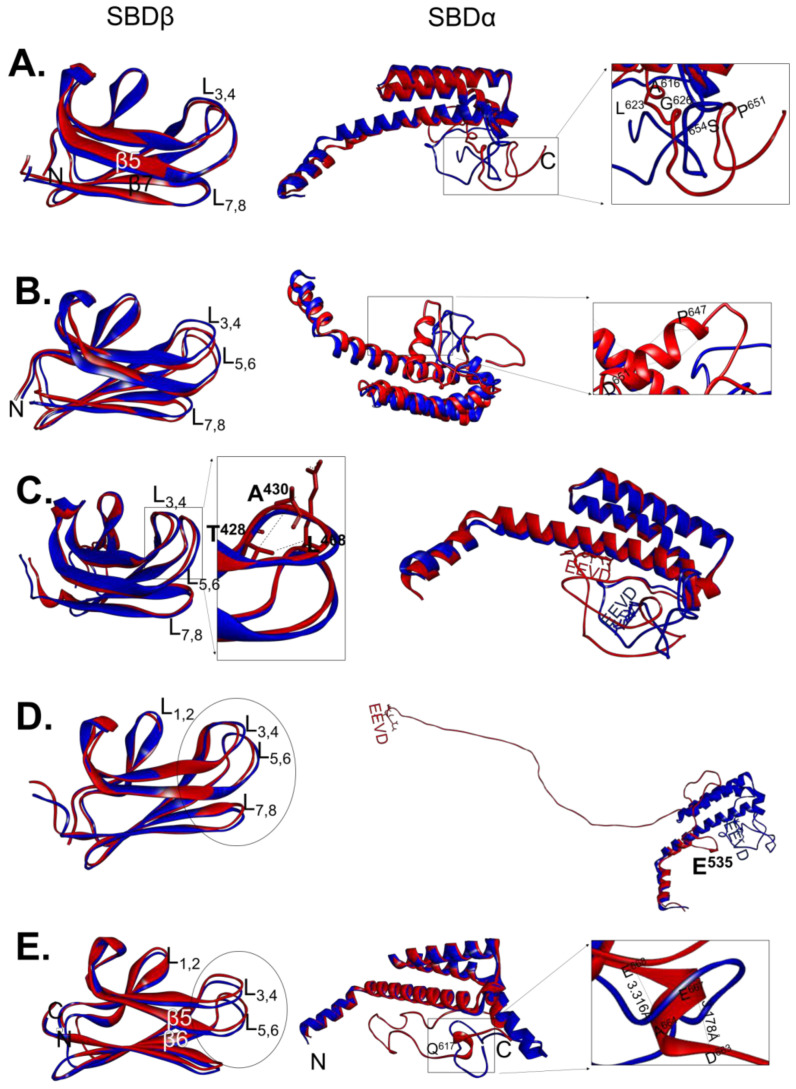
Superimposed three dimensional models of wild type SBDs of PfHsp70-1 and DnaK versus those of their GGMP variants. Superposition of the SBDβ and SBDα regions of PfHsp70-1 and DnaK against their respective GGMP variants was conducted: (**A**) PfHsp70-1 versus PfHsp70-1_G632_, the insert represents zoomed segment highlights H-bonding variations; (**B**) PfHsp70-1 versus PfHsp70-1_G648_, the zoomed SBD section with unique H-bonding is shown; (**C**) PfHsp70-1 and PfHsp70-1_G632-664_, with the insert showing unique H-bonding variations in SBDβ; (**D**) PfHsp70-1 versus PfHsp70-1_ΔG_ structures and highlighted are structural variations within the SBD; and (**E**) DnaK superposed with DnaK-_G_, insert highlights unique H-bonding pattern. The structures were visualized using Chimera.

**Figure 3 ijms-22-02226-f003:**
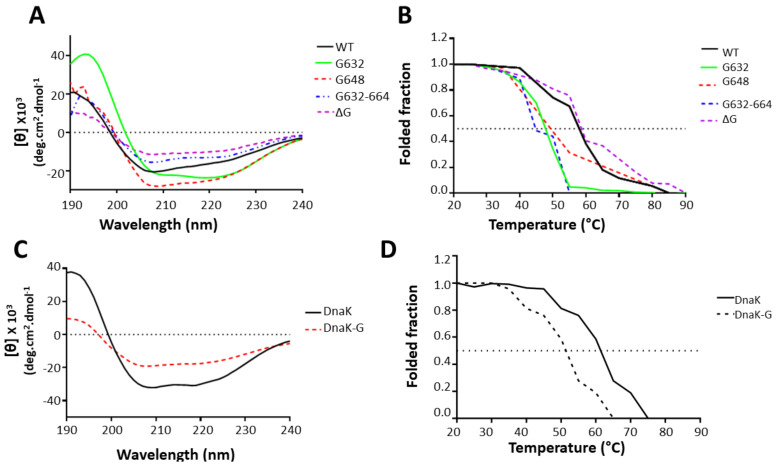
Secondary structure analysis. (**A**) The CD spectra of native PfHsp70-1, PfHsp70-1_G632_, PfHs70-1_G648_ PfHsp70-1_G632-664_, and PfHsp70-1_ΔG_ were presented as molar residue ellipticity (deg.cm^2^.dmol^−1^). (**B**) Temperature induced unfolding of the recombinant proteins was monitored by ellipticity measured at fixed wavelength of 222 nm as temperature was raised from 20 °C to 90 °C. Shown are: (**C**) CD spectra for DnaK and its mutant DnaK-_G_; and (**D**) folded fraction of DnaK and DnaK-_G_ monitored at 222 nm. Dotted line represents the melting temperature for 50% of the Hsp70 or its respective variants. Relative folded fraction of each protein was determined at a given temperature relative to fully folded state of the protein observed at 20 °C.

**Figure 4 ijms-22-02226-f004:**
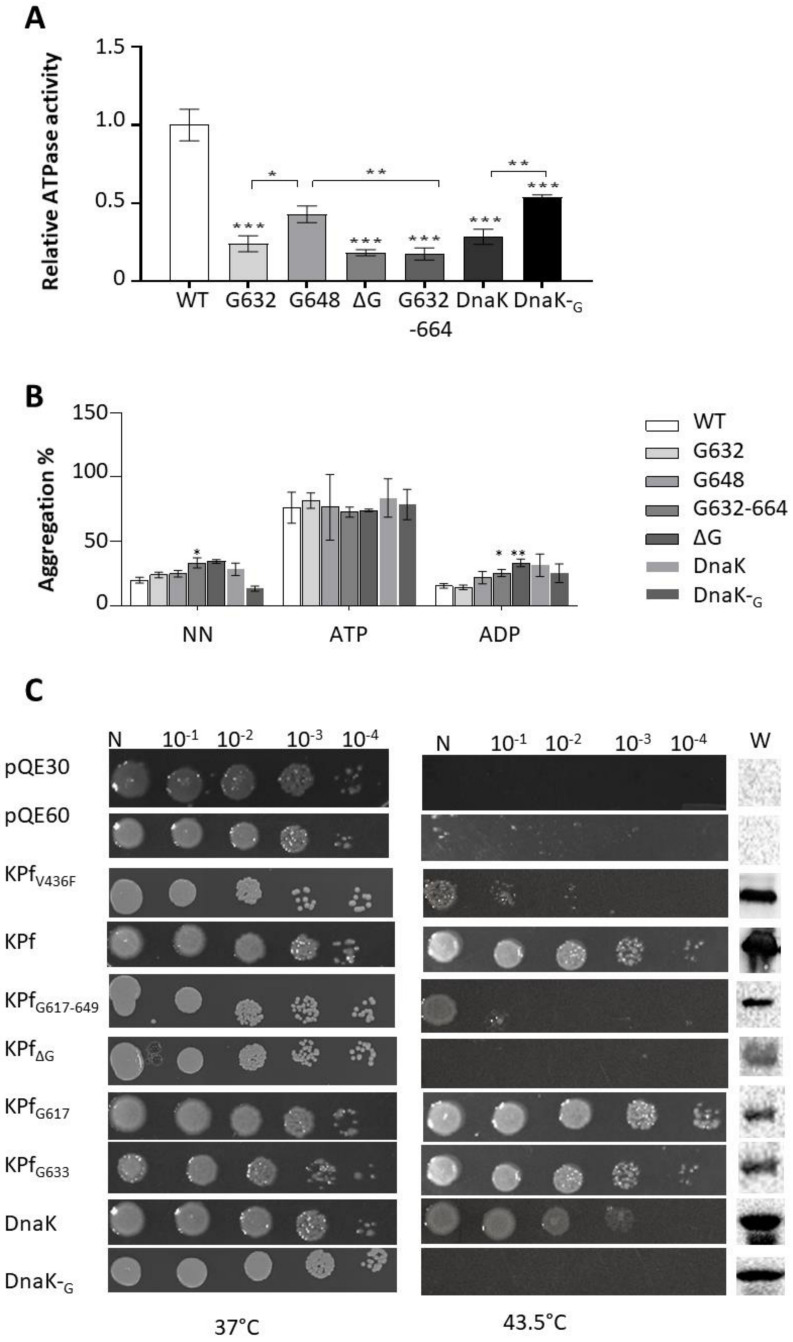
GGMP mutations compromise basal ATPase and chaperone functions of PfHsp70-1. (**A**) ATPase activity of the GGMP variants relative to PfHsp70-1 (WT). The basal ATPase activities of PfHsp70-1, DnaK, and their GGMP variants were determined by monitoring the amount of P*i* released determined by direct colorimetric readings conducted at 595 nm. (**B**) Heat induced aggregation suppression activities of PfHsp70-1 and its GGMP derivatives were monitored by exposing aggregation prone protein, MDH to heat stress at 51 °C in the presence of equimolar chaperone levels. The heat-induced aggregation of MDH was monitored spectroscopically at 360 nm. The assay was conducted in the absence of nucleotide (NN), or the presence of 5 mM ATP/ADP, respectively. (**C**) Complementation assay to determine the effect of the GGMP motifs on the function of chimeric protein, KPf, and DnaK. *E. coli dnaK756* cells transformed with plasmid constructs expressing either KPf, its GGMP mutants; DnaK, and its GGMP insertion mutant, DnaK-_G_, were incubated at the growth permissive temperature of 37 °C. Heat stress resilience of the cells was assessed by growing them 43.5 °C. Negative controls consisted of cells transformed with pQE60, pQE30 plasmid vectors and pQE60/KPf-V436F construct. Expression of the respective proteins was confirmed by Western blotting (“W”). Statistical analysis was carried out using one-way ANOVA at (* *p* < 0.05; ****
*p* < 0.01; *** *p* < 0.001).

**Figure 5 ijms-22-02226-f005:**
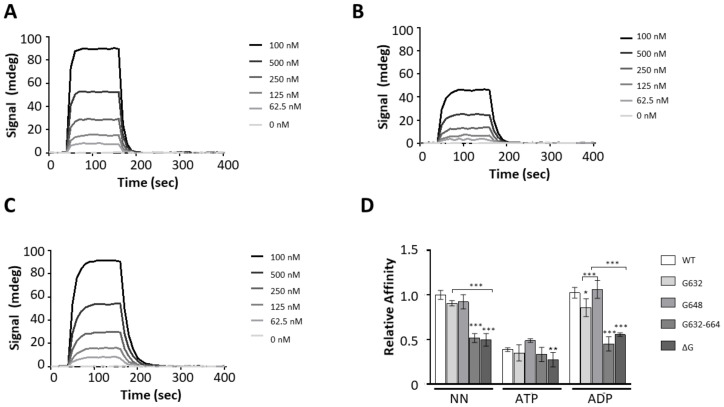
GGMP mutation compromises peptide binding. The representative SPR sensograms for the interaction of PfHsp70-1 GGMP variants with the peptide ANNNMYRR. Assay was conducted in the absence of nucleotides (**A**), and the presence of either 5 mM ATP (**B**) or 5 mM ADP (**C**), respectively. The relative affinities of PfHsp70-1 and its GGMP variants for ANNNMYRR were determined as shown (**D**). The error bars indicate data generated from three assays conducted using independent Hsp70 protein preparations. Statistical significance was determined by two-way ANOVA (* *p* < 0.05, *** p* < 0.01 and *** *p* < 0.001).

**Figure 6 ijms-22-02226-f006:**
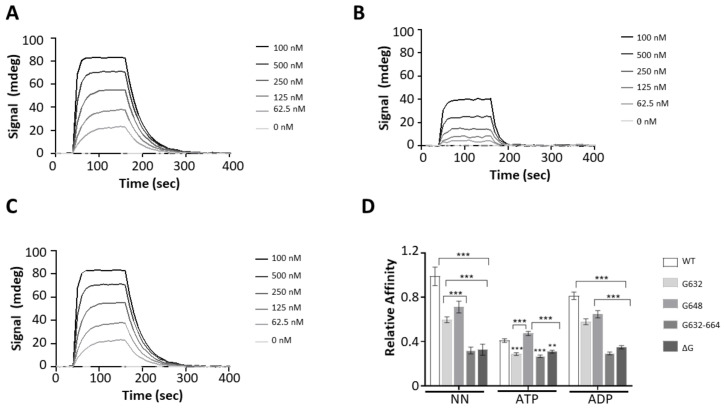
The GGMP motif of PfHsp70-1 modulates PfHop interaction. Representative SPR sensograms generated for assay monitoring interaction of PfHop and PfHsp70-1 GGMP variants are shown. Assay was conducted in the absence of nucleotides (**A**), and in the presence of either 5 mM ATP (**B**); or 5 mM ADP (**C**), respectively. The binding affinities of PfHop for PfHsp70-1 GGMP derivatives were normalized relative to the affinity of PfHop for wild type PfHsp70-1 as determined in the absence of nucleotides (**D**). The error bars represent data from three assays. Two-way ANOVA was used to determine the statistical significance at (** *p* < 0.01; *** *p* < 0.001).

**Figure 7 ijms-22-02226-f007:**
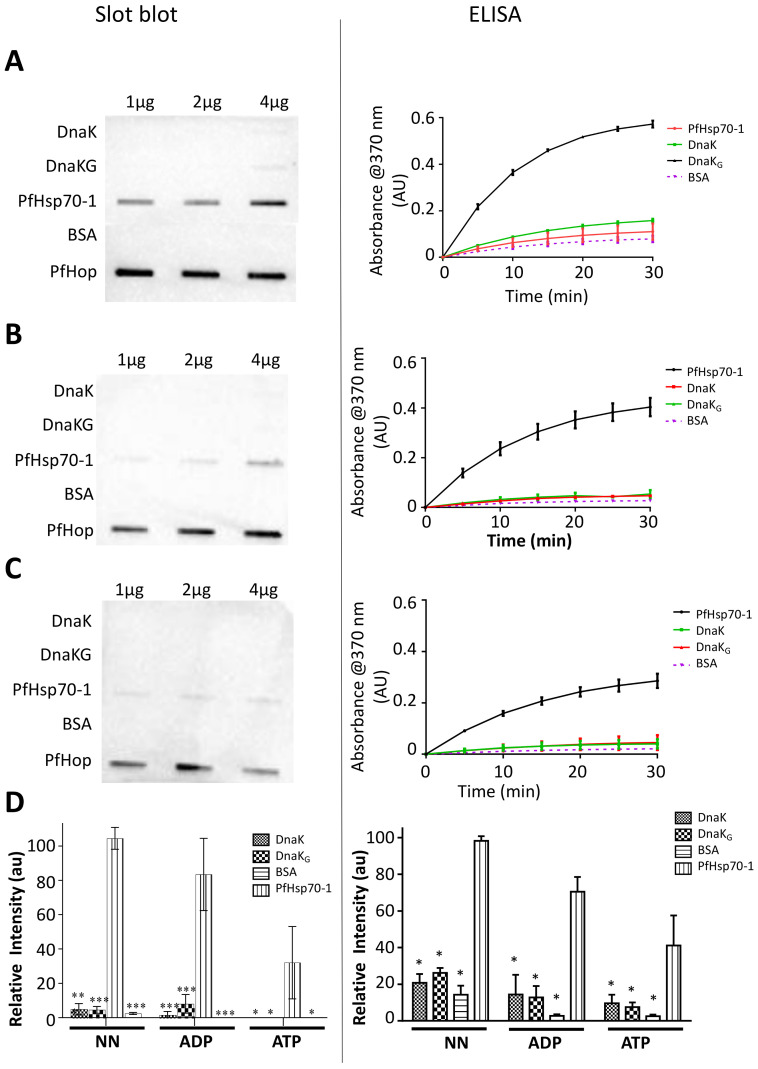
Insertion of GGMP repeat residues into DnaK did not result in PfHop binding. Slot blot and ELISA were conducted to explore interaction of DnaK-_G_ with PfHop. The slot blot images and graphs generated from ELISA absorbance readings to explore interaction of PfHop with DnaK and its variant, DnaK-_G_ in the absence of nucleotides (**A**), and in the presence of either 5 mM ATP (**B**), or 5 mM ADP (**C**), respectively. The relative intensities for PfHop-DnaK-_G_ interaction were normalized relative to intensity of signal obtained for DnaK-PfHop at their highest amounts of the proteins for the assay conducted in the absence of nucleotides (**D**). The error bars represent data from three assays. Two-way ANOVA was used to determine the statistical significance at (* *p* < 0.05; ** *p* < 0.01; *** *p* < 0.001).

**Figure 8 ijms-22-02226-f008:**
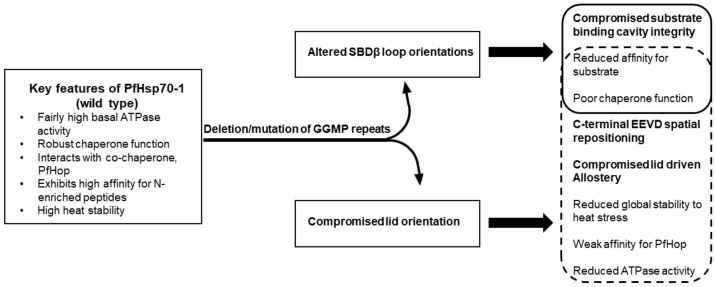
Model highlighting the roles of the GGMP repeat residues of PfHsp70-1. PfHsp70-1 exhibits key functional features which are compromised upon removal or mutation of the GGMP repeat residues. The main structural defects leading to compromised function appear to include reorientation of the SBDβ loops and the C-terminal lid region. Reorientation of the SBDβ loops and the lid both adversely impact on substrate affinity and chaperone function.

## Data Availability

Data is contained within the article or Appendix A.

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
