# Peer review of "Mutation of GGMP Repeat Segments of Plasmodium falciparum Hsp70-1 Compromises Chaperone Function and Hop Co-Chaperone Binding"

_ijms, 2021, doi:10.3390/ijms22042226_

Round 1

Reviewer 1 Report

In their m/s Makumire et al report the data on the probable role of GGMP peptides in the structure and function of Hsp70 chaperone from Plasmodium falciparum. The article is thought to establish a link between intimate features of substrate-binding domain of Hsp70 and its chaperonic capacity.

The authors did a big work on constructing PfHsp70 and DnaK mutants and characterizing them. However, the article is hard to read because it is overloaded with data and the text is full of repeats and of unnecessary details some of which should be transferred to the other parts, most probably to Discussion and to Supplement.

Major concerns.

-Describing separate features of PfHsp70 mutants authors in the end of each section or even paragraph conclude that GGMP segments are important for PfHsp70 function and their relevance depends on their position on the protein molecule. However, there is no particular predominance of a certain GGMP motif in a distinct functional feature of the protein. Moreover, deletion of the whole segment with GGMPs has lesser effect on the ATP-ase activity or anti-aggregation capacity than that of separate “point” mutations. Can the authors basing on their data explain the mechanism of the involvement of GGMP at least in one particular function of Hsp70, target peptide binding regulated by ATP (in presence or absence of ATP)?

-The authors mentioned that GGMPs are present in molecule of human Hsc70, so can they check whether these two proteins give similar results in assays employed in this work, especially target peptide binding and anti-aggregation? Certainly, it could be fantastic to compare more samples of Hsp70 with distinct amount and spatial positions of GGMP in above settings!

-I am not sure whether viability of E. coli is a good way to measure the chaperonic capability of Hsp70 because of great difference between procaryotic and eucaryotic organisms in the cellular protein environment. Therefore to compare PfHsp70 I would choose yeast or Drosophila proteins instead of DnaK. Instead of viability test one can employ refolding chaperonic assay with thermodenaturated luciferase. I hope this experiment gives more proofs to statements of this m/s.

Minor.

The text needs to be shortened and multiple duplications of phrases and statements, especially relating to conclusions and suggestions should be removed from Results. The Discussion part may be also shortened; a scheme depicting a possible role of GMMP in Hsp70-driven chaperonic activity is appreciated. 

Author Response

We appreciate feedback from reviewer 1 which we attended to in the attached document with our point by point rebuttals inserted. To make it easier to follow, our comments are in bold. 

Reviewer 2 Report

The authors performed several experiments to define the possible roles of GGMP repeats in the regulation of Hsp70-associated chaperone function, substrate binding, and PfHsp70-PfHop interaction by analyzing the effects of mutations in the repeats or insertion of the repeats into DnaK. Biochemical, biophysical, and complementation assays were performed, and 3-D models constructed from these observations were analyzed. The authors demonstrate that GGMP repeats maintain protein stability and play an important role in ATPase activity, chaperone function, and substrate binding of PfHsp70.

The background section presents the current knowledge on heat shock proteins, notably those of malaria parasites. The materials and methods section describes in detail the methods used in this paper. The results are presented in detail, both in the main text and in supplementary materials. Some parts of the Discussion are repetitions of the Results section. These can be deleted.

There is no major problem with English, but punctuations should be re-checked, and species names in Latin should be italicized throughtout the text. References should be in the same format.

The paper is clear and very well written. Minor corrections would improve this manuscript.

Major comments:

none

Minor comments:

Line 30, Abstract: The authors should add a short conclusion (1 or 2 sentences) at the end of the abstract.

Lines 35, 88: P. falciparum (instead of Plasmodium falciparum, first written in its complete form in line 34).

Line 109, also line 366: Apicomplexa (not “Apicocomplexa”); Apicomplexa is a phylum, not a “kingdom.”

Lines 111-112, 113, 115, 120: Cryptosporidium parvum (in italics; spelling of Cryptosporidium), Trypanosoma cruzi (in italics), Plasmodium berghei (in italics), Toxoplasma gondii (in italics); Plasmodium (in italics); P. berghei, P. vivax; Saccharomyces cerevisiae

Line 112: close parenthesis (Figure 1A)

Line 118: delete the comma after “remains”

Line 122: delete one “the” in the phrase “the the GGMP motif…”

Lines 123, 471, “Apicomplexan species”: I would say either species belonging to Apicomplexa phylum or apicomplexans.

Figure 1 A: All Latin names should be in italics, as with Plasmodium falciparum. Spelling: Trypanosoma brucei. Please also check the figure legend. Line 128: What does “MSA” stand for?

Line 293, “…account for its this phenomenon”: delete “its”

Line 295: delete the comma after DnaK

Lines 299-302, “The basal ATPase activities…was conducted…” suggest “activities…were determined by monitoring the amount of Pi released measured by direct colorimetric readings…”; “Heat induced aggregation suppression activities… was monitored…” suggest “were monitored…”

Line 312, “statistical analysis was carried out using one-way ANOVA”: This sentence should be at the end of the materials and methods section. Same remark for other figure legends.

Line 329, “not statistically significant”: Please add the P-value (P > 0.05)?

Line 334, 385: “apo” in italics

Line 344, “in vitro degenerated data”: Should it be in vitro “generated” data?

Line 439: Data show that

Line 443: densitometric analyses…were used

Lines 448-449: “It is possible that the presence of … DOES not promote….”

Line 481: Does the expression “GGMP repeats were more enhanced” mean that there is a higher number of repeats in T. gondii? Please clarify.

Line 489: delete the comma after “generated”

Line 494, “furthermore, introduced GGMP mutations in KPf…”: Furthermore, WE? introduced GGMP mutations

Line 552, “yeast Hsp70 bind”: binding site?

Line 630: delete the comma after “study”

Lines 649-651: close parenthesis after [43])

Lines 666-667: “expression and purification…were conducted…”

Line 675, 680, 683: imidazole (small letter “i”)

Line 678, 6 000 ×g

Line 687, CD: This abbreviation was used earlier in the text. “Circular dichroism (CD)” should be presented the first time this term is used.

Line 716, “reaction mix at to achieve…” delete “at”

Line 747, “ELISA were conducted”: ELISA was conducted

Line 766, new sub-section 4.12: I suggest that the authors add “statistical analysis” sub-section here and describe the statistical tests used (ANOVA).

Ref 5: gallate (small letter “g”)

Ref 6: plasmodial (small letter “p”)

Please add DOI: ref 8, 9, 11, 12, 14 and other references.

Ref 10: delete the opening parenthesis in the title “(“

Ref 11, 12: The format of the author names is different from that of other references.

Ref 13: E. coli in italics

Ref 43: decarboxylase (small letter “d”)

Ref 46: Malar J 2010;9:236.

Please check that all references are cited in the text. Please also write in full what the abbreviations stand for when they are first used in the text.

Author Response

We appreciate the feedback from the reviewer. We have attached a document with our rebuttals. The comments have greatly improved the quality of the MS. Note, page references are based on the m/s version with tracked changes shown.

Round 2

Reviewer 1 Report

The m/s in its present form looks more appropriate though the authors do not respond to certain comments because of epidemic situation. My remaining concern is that I still do not understand why the authors focused on the interaction of Hsp70  with HOP but not with a DNAJ class co-chaperone because particularly the latter and Bag class proteins execute all functions related to chaperonic mechanism governed by Hsp70s. However, this comment does not influence generally positive estimate of the work.